# Costa Rican Propolis Chemical Compositions: Nemorosone Found to Be Present in an Exclusive Geographical Zone

**DOI:** 10.3390/molecules28207081

**Published:** 2023-10-14

**Authors:** Eduardo Umaña, Godofredo Solano, Gabriel Zamora, Giselle Tamayo-Castillo

**Affiliations:** 1Centro de Investigaciones en Productos Naturales (CIPRONA), Universidad de Costa Rica, San José 11501, Costa Rica; eduardo.umana.rojas@una.ac.cr (E.U.); godofredo.solano@ucr.ac.cr (G.S.); 2Centro de Investigaciones Apícolas Tropicales (CINAT), Universidad Nacional, Heredia 3000, Costa Rica; luis.zamora.fallas@una.ac.cr; 3Escuela de Química, Universidad de Costa Rica, San José 11501, Costa Rica

**Keywords:** propolis, nemorosone, chemical fingerprinting, NMR

## Abstract

Background: The chemistry of Costa Rican propolis from *Apis mellifera* remains underexplored despite its potential applications. This study identified its chemical composition, linking chemotypes to antioxidant potential. Methods: Proton nuclear magnetic resonance (^1^H NMR) spectra were obtained for 119 propolis extracts and analyzed using multivariate analyses. In parallel, 2,2-diphenyl-1-picrylhydrazyl (DPPH) radical scavenging assay was used to assess antioxidant activity. A generalized linear regression model (GLM) correlated this with its chemical profiles and geographical origin. Chromatographic methods were used to isolate active and inactive compounds, which were identified using nuclear magnetic resonance (NMR) and high-resolution mass spectrometry (HRMS). Results: Principal component analysis (PCA) revealed three chemical profile groups for the 119 propolis extracts, explaining 73% of the total variance with two components. Radical scavenging activity was found to correlate with chemical composition. Isolation yielded *n*-coniferyl benzoate in type I (EC_50_ = 190 µg/mL, ORAC = 0.60 µmol TE/µmol) and nemorosone in type II (EC_50_ = 300 µg/mL, ORAC = 0.7 µmol TE/µmol). Type III was represented in terpene-like components, which exhibited lower antioxidant activity. Conclusions: This study categorizes Costa Rican propolis into three chemical types and identifies two key components linked to antioxidant activity. Notably, nemorosone, a valuable natural product, was found to be highly concentrated in a particular region of Costa Rica.

## 1. Introduction

An informal survey conducted among stores in Costa Rica, specializing in selling organic products, revealed a variety of *Apis mellifera* propolis-based products available in different presentations. Developing secondary products derived from propolis is promising, given that beekeeping economic activities have suffered losses. However, there is a lack of research on the chemical composition within the country. As a result, these products are being sold without proper health registration or may include dietary claims linked to primary accompanying materials such as honey, ginger, and vitamins.

The chemical composition varies significantly based on its geographic origin [1,2]. For instance, European propolis is primarily characterized by the presence of flavonoids and phenolic acids and their esters [3]. Brazilian green propolis, on the other hand, consists of prenylated coumaric acid and acetophenone derivatives [4,5,6]. Cuban red propolis primarily consists of isoflavonoids, while brown propolis is composed mainly of prenylated benzophenones [7,8]. Furthermore, Central American propolis from Honduras and El Salvador have shown the presence of cinnamic acid derivatives, flavanones, chalcones, and aromatic acids [9,10].

Propolis has demonstrated a range of beneficial properties, including antibacterial (specially against Gram-positive bacteria) [11], antifungal [12], antiproliferative and immunomodulatory [13,14,15], and antioxidant activity [16,17].

The challenge in developing propolis-based products lies in their standardization [2], a task that becomes intricate in countries with abundant biodiversity [18,19]. As a matter of fact, propolis from different regions of Chile have specific botanical origins of resins from *Sorghum bicolor*, *Lotus* sp., *Acacia* sp., *Pinus vadiata*, *Eucalyptus* sp., *Salix babylonica*, and *Quillaja Saponaria* [20]. Given that *Apis mellifera* bees gather nectar and pollen from the surrounding flora within a radius of approximately 3 km and considering that Costa Rica stands as one of the world’s biodiversity hotspots, with around 12,000 terrestrial plant species [21] in just 51,100 km^2^, it is possible that propolis compositions vary between regions and over time. In the absence of a comprehensive study on the chemical composition of Costa Rican propolis, this hypothesis necessitates validation.

Furthermore, prior to harnessing propolis and undertaking product development, it is a prerequisite to acquire information on the chemical composition and undertake spatiotemporal studies of the raw material. Once this information is established, the identification of a chemical marker can subsequently facilitate its quantification. For the structural assignment of complex matrices, such as propolis’s extracts, hyphenated mass chromatography methods have been used previously [7,22,23]; however, these techniques have an important limitation when describing the chemical nature of raw materials for the first time: the components must be ionizable and are thus dependent on the ionization source used. They must also to be compatible with the chromatographic technique employed. In this context, the use of nuclear magnetic resonance (NMR) emerges as the optimal approach for analyzing complex matrices for the first time. While it has limitations in terms of sensitivity, it excels in providing a comprehensive overview of the chemotype and relative concentration of all components dissolved in a solvent. 

Attempts have been reported to analyze and standardize propolis extracts according to their origin using NMR and multivariate analysis. Watson et al. [24] applied a principal component analysis (PCA) on ^1^H NMR propolis fingerprints of extracts and concluded that the resulting chemical space correlated well with their geographical origin. The authors also found a correlation between the chemical space and the antioxidant activities measured. Other works [25,26,27,28] account for the use of these statistical tools to correlate geographical origin, chemical composition, and biological activities. On the other hand, Bertelli et al. [26] described the use of ^1^H NMR experiments to obtain a chemical fingerprint and identify common phenolic components in mixtures as a tool to analyze and standardize propolis extracts.

We hypothesize that, given the abundant plant biodiversity in Costa Rica, propolis produced by *Apis mellifera* bees may exhibit different contents according to their geographical origin and hence may be difficult to standardize and require more in-depth study. Furthermore, while the biological activities of Costa Rican propolis remain unexplored, the anticipated chemical diversity suggests potential variations in their biological properties as well. 

In accordance with the previous statement, our objectives encompassed the following: (1) employ ^1^H NMR spectroscopy to systematically categorize the chemical compositions of extracts of propolis from a substantial representation of Costa Rican apiaries; (2) correlate, if possible, the chemical information with antioxidant activity using the conventional assay of 2,2-diphenyl-1-picrylhydrazyl (DPPH); and (3) subsequent to this comparative analysis, isolate the specific compounds responsible for the antioxidant activity to characterize them chemically and determine their antioxidant potential with the oxygen radical absorbance capacity (ORAC) test.

## 2. Results

### 2.1. Chemical Spaces of Costa Rican Propolis

#### 2.1.1. Propolis Sampling

The sampling strategy involved random selection without substitution, resulting in the distribution pattern illustrated in Figure 1, which encompassed a total of 119 samples of propolis from 24 selected apiaries, equivalent to 2.9% of all Costa Rican apiaries. Notably, the sampling was biased towards regions with a higher concentration of apiarian producers, i.e., the Northern Pacific and the central regions of the country. Interestingly, no apiaries were identified in the Caribbean region.

In this study, we collected five samples from each chosen apiary, and each sample was sequentially labeled with a distinct number. Additionally, to ensure biological replication, each of these five samples from a given apiary was further labeled with a unique letter.

Costa Rica is characterized by high elevations, which divides the country into several slopes: one towards the Caribbean, the second towards the Pacific coast and the third one towards the North. Therefore, it is possible to find apiaries at altitudes as high as 1700 and others as low as 50 m above sea level (masl), with a vast majority located below 1000 masl.

#### 2.1.2. NMR Measurements and Data Treatment

In our study, a universal extraction with methanol was performed; extracts were kept overnight in a freezer to allow for the precipitation of waxes and centrifuged the following day. An internal standard (maleic acid, one singlet at 6.30 ppm) was added to an exact fraction of the dried remnant (6.0 ± 0.1 mg) in a Shigemi NMR tube, and then its ^1^H NMR spectrum was measured using typical metabolomic experimental conditions. The addition of this internal standard allowed for the alignment and normalization of data, as an additional option for the PCA analysis [27,28]. Figure 2C represents the three most common ^1^H NMR obtained.

#### 2.1.3. Statistical Analysis

Figure 2A shows a score plot of the PCA scores of the 24 apiaries selected for this analysis. The analysis using three components resulted in a higher explained variability (83% vs. 73%); however, the tendencies were the same. Consequently, the analysis was based on PC1and PC2. Most of Guanacaste samples (55%) were distributed along the second component, accounting for 23% of the total data variability. These samples were categorized as members of the Chemotype I group. As can be seen in Figure 2B, samples from South Pacific regions, the Central Pacific, the North Zone, and the South of Central Valley are mainly represented by the PC1 component, which represents 51% of spectral data variability and was categorized as part of the type II group. The remaining samples were clustered in a chemical space defined by negative scores regarding the PC1 and PC2 components (Figure 2A, samples clustered at the left bottom in yellow and orange) and categorized as type III. This cluster primarily consisted of samples from the Central Valley (north and south sub-regions), Guanacaste, the Pacific Coast, and the central region.

These results suggested that Costa Rican propolis could be grouped into three chemotypes, a classification that was confirmed through self-organizing map (SOM) analysis. The first chemotype (I) was present in 20.2% of samples, the second one (II) in 29.4%, and the third (III) in 50.4%. A geographic distribution trend of samples in the resulting chemical types was observed. In Figure 1, these chemotypes are represented by primary colors: yellow for type I, red for type II, and blue for type III, identifying apiaries where only one predominant chemotype was present. Combinations of colors (secondary colors) represent apiaries where more than one chemotype was found. For example, Apiary 10 is colored orange since four samples of Chemotype I (yellow) and one of Chemotype II (red) were found here.

### 2.2. Isolation and Identification of Major Compounds

Upon conducting the preparative fractionation of representative propolis extracts of each chemotype, using vacuum liquid chromatography, preparative thin-layer chromatography, and semi-preparative high performance liquid chromatography, we isolated four compounds, which were characterized by 1D and 2D NMR techniques and HRESIMS (see Appendix A). 

The major components were *n*-coniferyl benzoate and its *O*-methyl derivative, nemorosone, and agathadiol. The first and third compounds from Chemotypes I and II showed antioxidant activities, while the last one from Chemotype III showed no antioxidant activity. Loadings defining PC1 and PC2 (from the PCA) were primarily associated with bins at 1.65 and 3.85 ppm, as shown in Figure 3B,D. These bins were linked to nemorosone (Figure 3A,B) and *n*-coniferyl benzoate (Figure 3C,D). The boxplots in Figure 3A,C show the normalized intensity of bins between 1.57 and 1.69 ppm and between 3.81 and 3.89 ppm, respectively, with respect to their apiary origin. The prevalence of nemorosone in apiaries from the southern region (highlighted in pink), and of *n*-coniferyl benzoate in Guanacaste (northeast, in green) were remarkable and highly unexpected. To support this observation, an additional PCA analysis was conducted, including 99 propolis samples and based solely on the chemical shifts regarding nemorosone and *n*-coniferyl benzoate. The same clustering pattern as was observed in Figure 2A,B persisted, as shown in the Appendix A. A linear discriminant analysis (DA) model was constructed to explain these observations; the success rate was 100% and the predictive potential through cross-validation was in the order of 92.9% (see Appendix A). These results demonstrated that the ^1^H NMR spectra of Chemotype I propolis were primarily characterized by *n*-coniferyl benzoate signals, whereas those from Chemotype II were predominantly characterized by nemorosone. 

#### 2.2.1. *n*-Coniferyl Benzoate

Compound **1** was isolated using vacuum liquid chromatography from extract 6 A, a representative propolis of Chemotype I collected in Nandayure, Guanacaste. The protonic spectrum clearly showed two aromatic rings, one monosubstituted and the other one with a pattern of 1, 2, 4. Additionally, it presented a vinylic system at 6.69 and 6.30 ppm with a coupling constant of 16.0 Hz, typical of protons in a *trans* geometry. These spectroscopic facts were consistent with a coniferyl fragment with two oxygenated groups, one of them appearing as an *O*-methylated group with a singlet of 3.87 ppm. The MS spectrum showed the molecular formula C_17_H_16_O_4_, detected as [M − H]^−^ (calc = 283.0970; obs = 283.0968). The structure of *n*-coniferyl benzoate was further confirmed through comparison with the ^1^H NMR and ^13^C NMR data available in the literature [29]. The compound was found to exhibit antioxidant activity through both DPPH and ORAC tests (see Table 1).

#### 2.2.2. Nemorosone

Compound **2** was isolated through a combination of preparative thin-layer chromatography and high-performance liquid chromatography of an extract obtained from propolis Chemotype II (sample 19 C), from Coto Brus, a locality found in the southern part of Costa Rica. The ^1^H NMR spectrum revealed the presence of three isopentenyl fragments attributed to typical proton signals on carbon double bonds at 5.24 ppm, 5.04 ppm, and 5.06 ppm and six methyl groups at 1.67 ppm, 1.66 ppm, 1.62 ppm (9 H), and 1.57 ppm. The structure of a bicycle [3.3.1] nonane 2,4,9-trione was deduced from chemical shifts in the ^13^C NMR spectrum at 79.5 ppm, 63.7 ppm, and 47.4 ppm corresponding to the bicycle core and three conjugated ketonic carbonyls at 187.2 ppm, 189.7 ppm, and 197.9 ppm. The former two carbonyls were in keto-enolic equilibrium with an olefinic carbon at 118.4 ppm. The position of isopentenyl moieties was established via two-dimensional NMR experiments. 

The presence of a benzoyl moiety was deduced from the aromatic signals of the ^1^H NMR spectrum that were integrated for five protons and the presence of a carbonyl at 214.4 ppm. The molecular weight for Compound **2** was consistent with the molecular formula C_33_H_42_O_4_, detected as [M − H]^−^ (calculated = 501.3005, observed = 501.3000, 1.0 ppm error), and [M + H]^+^ (calculated = 503.3161, observed = 503.3149, 2.4 ppm error). The structure of Compound **2** was suggested to be nemorosone, and this was confirmed by further comparison to data from the literature [30]. The compound exhibited antioxidant activity in both DPPH and ORAC tests (see Table 1).

### 2.3. Isolation and Characterization of Inactive Components

#### 2.3.1. Agathadiol

Compound **3** was isolated by applying a combination of PTLC and HPLC techniques to a propolis sample from Turrubares, located in the Pacific Central region of Costa Rican, that represented Chemotype III. The ^1^H NMR spectrum showed the absence of aromatic signals, and the ^13^C NMR presented twenty chemically different carbons and an absence of carbonyl moieties, suggesting that the molecule might be a diterpene. The signals at 3.73 ppm, 3.26 ppm, and 4.08 ppm were consistent with a diterpene of a labdanediol skeleton. The structure was proposed to be agathadiol, and this was confirmed via a comparison with the literature data [31]. This compound was inactive in the DPPH test.

**Table 1 molecules-28-07081-t001:** Measured antioxidant activities of crude extracts (mean value) and isolated active compounds *.

Sample	EC_50_ DPPH (µg/mL)	ORAC (µmolTE/µmol)	ORAC (µmolTE/mg)
Extracts Apiar 6 (Chemotype I)	58 ± 19	Not apply	1.8 ± 0.2
Extracts Apiar 19 (Chemotype II)	56 ± 13	Not apply	1.7 ± 0.2
Extract Chemotype III	868 ± 54	Not apply	<0.1
*n*-Coniferyl benzoate 1	190 ± 10	0.60 ± 0.06	2.1 ± 0.2
Nemorosone 2	300 ± 15	0.70 ± 0.2	1.4 ± 0.4

* For *n*-coniferyl benzoate, no literature data were found. For nemorosone, Cuesta-Rubio et al. [32] reported DPPH activity (EC_50_) of 22.2 µg/mL at 37 °C.

#### 2.3.2. *O*-Methyl-*n*-coniferyl Benzoate

In addition to isolating *n*-coniferyl benzoate, its *O*-methylated derivative was also isolated. The proton NMR spectrum clearly showed two aromatic rings, one monosubstituted and another with a 1,2,4 pattern. Additionally, a vinylic system was observed at 6.71 ppm and 6.34 ppm, showing a *trans* coupling constant (*J* = 15.6 Hz) consistent with the characteristics of a coniferyl benzoate derivative. Finally, two methoxy groups were observed at 3.85 ppm and 3.87 ppm, suggesting that this compound was indeed the *O*-methylated derivate of *n*-coniferyl benzoate. The structure was confirmed via two-dimensional NMR experiments. Due to the structural modification of its catechol moiety, this compound exhibited no antioxidant activity in the DPPH test.

### 2.4. Antioxidant Activity of Costa Rican Propolis Extracts

Figure 4A displays PCA score plots for each extract, incorporating information about their respective effective concentration (in µg/mL) that inhibits DPPH by 50%. In the plot, smaller circles represent extracts with higher antioxidant activity. Additionally, geographical origins are represented with different colors. The most active samples predominantly originated from the South of Costa Rica, and some from the Guanacaste region. Notably, the most active samples aligned with both PC1 and PC2 in the plot and corresponded to the chemical spaces occupied by propolis extracts of Chemotypes I and II, respectively. Conversely, extracts with lower potency are closer to the type III chemotype. In summary, these results demonstrate that the most effective antioxidant extracts were associated with Chemotypes I and II.

To evaluate whether this observation was statistically valid, we ran a generalized linear regression model (GLM). Given that there was no normal distribution in EC_50_ values and that the data from our variables were presented in decimals, a GAMMA model was used. The variance inflation factors (VIFs) obtained for PC1 (1.22) and PC2 (1.22) showed that the chemical fingerprint had an effect on the obtained EC_50_, and the model did not show any co-linear problems. Figure 4B,C shows the significant correlations between the antioxidant activity and chemical composition of propolis extracts. The correlations exhibited a decreasing pattern along the axes, which suggests a non-linear and diminishing relationship between the variables and the principal components. This highlights the importance of the early variables in explaining the variance (R^2^ = 0.6465) and structure in the dataset.

Furthermore, a PLS model with three components demonstrated a correlation between propolis samples exhibiting an EC_50_ of less than 1000 µg/mL and the chemical fingerprint derived from the ^1^H NMR analysis. This model yielded a R^2^ value of 0.74 and a predictive R^2^ of 0.62, indicating its robustness and quality. Given that the chemical fingerprints of Chemotypes I and II were predominantly characterized by the ^1^H NMR signals of *n*-coniferyl benzoate and nemorosone, respectively, it was inferred that the EC_50_ of these propolis extracts also exhibited a strong correlation with these specific components. 

Notably, the propolis extracts that displayed the most EC_50_ DPPH activity were found to display more intense signals at 7.36 ppm (r = 0.713, *p* < 0.0001) and 7.52 ppm (r = 0.768, *p* < 0.0001), which correspond to nemorosone and *n*-coniferyl benzoate, respectively. This relationship was further supported by bioassay-guided fractionation of extracts 6A, 19C, and 23C, employing DPPH-TLC detection to identify active components. Additionally, the data from pure compound analysis indicated that *n*-coniferyl benzoate exhibited superior antioxidant activity compared to nemorosone, as was evident in Table 1. Finally, it is remarkable that the extracts displayed higher antioxidant activity than the individual main components, as will be discussed in more detail later.

## 3. Discussion

Our study revealed that Costa Rican bee propolis are distributed across the country in three distinct chemical spaces, delineated by their ^1^H NMR spectra, obtained through PCA analysis. Samples of Chemotype I were notably set apart from others along the PC2 component, displaying positive scores. Similarly, samples of Chemotype II were characterized by positive scores along the PC1 component. On the other hand, the remaining samples were situated in a space defined by negative values for both components, denoted as Chemotype III. 

Despite the high plant diversity reported for Costa Rica, it was striking to find that 100% of samples from the South were composed of mainly nemorosone and categorized as Chemotype II. On the other hand, a vast number of samples from the Central Valley (north and south sub-regions, 67.5%) and Central Pacific (60%) regions were identified as Chemotype III. Nearly half of samples from the Guanacaste province were placed in the Chemotype I (45%) category while the rest were identified as Chemotype III (51.3%). A chi-square test for association revealed a significant dependence (*p* < 0.0001) between geographic zone and the chemical propolis type found within an apiary. Each pair of data sets’ chi-square contributions to the overall chi-square value (Pearson chi-square value of 73.75) highlighted notable associations. The most pronounced relations included the exclusive occurrence of Chemotype II in southern Costa Rica (25.9118) and the high percentage of Chemotype I (13.0579) in Guanacaste, with the contrasting minimal presence of Chemotype II (9.5578) in the same region.

Through our assessment of antioxidant activity using the DPPH test, a congruence emerged between extracts demonstrating higher activity and the chemical spaces corresponding to Chemotypes I and II. Whether a propolis extract sample belonged to one specific chemotype was determined by the chemical fingerprint and substantiated by the relative intensities of the ^1^H NMR signals. Ultimately, this classification relied on the extract’s relative concentration of *n*-coniferyl benzoate and nemorosone.

As previously mentioned, the strongest antioxidant activity was observed in extracts showing heightened intensity signals at 7.52 ppm (r = −0.768, *p* < 0.0001) and 7.36 ppm (r = −0.713, *p* < 0.0001), which correspond to *n*-coniferyl benzoate and nemorosone, respectively. This indicates that these compounds significantly contributed to the extract’s overall antioxidant activity, carrying substantial individual influence. These findings implied that Costa Rican propolis extracts enriched with either of these components in higher concentrations were likely to exhibit superior antioxidant activity compared to those with lower concentrations. Essentially, both compounds emerge as robust chemical markers for assessing the exceptional antioxidant performance of high-quality Costa Rican propolis.

The nemorosone activity results from Cuesta-Rubio et al. [32] showed an almost 14-fold increase in DPPH EC_50_. We attribute the difference between the reported data and ours to the difference in temperature used in the experiments (37 °C for theirs, 23 °C for ours). Schaich et al. [33] suggested that the DPPH reaction rate heavily depended on the antioxidant’s efficacy in sterically hindering the radical center, implying that factors boosting effective molecular collisions between reagents, such as temperature, could enhance antioxidant potency. Indeed, our nemorosone results, obtained at room temperature, were superior to those reported for 7-epi-clusianone (EC_50_ value of >800 µg/mL [34], a regioisomer isolated from *Garcinia brasiliensis*).

Interestingly, some coniferyl derivatives like ferulic acid and coniferyl alcohol have been reported to exhibit slightly elevated antioxidant activities at 37 °C (149 µg/mL and 146 µg/mL, respectively) compared to *n*-coniferyl benzoate measured at 23 °C [35]. However, this impact did not show a consistent relationship with temperature difference. The influence of temperature on antioxidant reactivity with the DPPH radical varied among different compound families, primarily due to differences in molecule size. For larger molecules like nemorosone, which react slowly (as observed in the TLC spray test where the pink color emerged 2 min after applying the DPPH methanolic solution), temperature had a significant effect on EC_50_ by increasing the number of effective molecular collisions with the DPPH radical. This differed from smaller phenolic acid derivatives that react more swiftly (yellow color appears immediately after spraying the DPPH solution). Furthermore, our analysis highlights the comparable antioxidant capacity of Compounds **1** and **2** to that of vitamin E (Trolox in its water-soluble form), as evidenced by ORAC test results.

Table 1 reveals an intriguing observation: the antioxidant activity of the extracts, as measured by the DPPH test, was higher compared to that of pure compounds. This contrast needs further investigation. The GML model’s correlation, which connects DPPH activity with the composition of propolis matrixes assessed via ^1^H NMR analysis, is thought-provoking. One possible explanation for this contrast is the presence of additional antioxidant compounds whose ^1^H NMR signals overlapped with those of the main compounds. Alternatively, it could stem from a synergistic effect between certain matrix components, enhancing the potency of the crude extract. Furthermore, the ORAC values obtained for both extracts and pure compounds suggest a common underlying mechanism. The active compounds in the extracts were present at concentrations not exceeding 50% mass/mass (data to be published soon).

Regardless of the specifics, Costa Rican bee propolis extracts with higher concentrations of *n*-coniferyl benzoate and nemorosone exhibited superior antioxidant activity when compared to those with much lower concentrations of these two compounds, which were predominantly represented by the Chemotype III (rich in terpene-like compounds, such as agathadiol).

However, our findings revealed that the individual antioxidant activities of nemorosone and *n*-coniferyl benzoate were significantly lower than anticipated. For instance, *n*-coniferyl benzoate accounted for only 30% of the total antioxidant activity of the extract from which it was isolated, and nemorosone contributed only 19%. Consequently, based on these results, we concluded that both chemical markers were of a qualitative rather than quantitative nature. The prevalence of one of these markers in a propolis sample signified the relative dominance of its specific botanical source over others. The resin must have been composed of additional substances that exerted an antioxidant effect, either individually or synergistically, enhancing the overall antioxidant activity compared to samples of Chemotype III. 

These results underscore the effectiveness of using ^1^H NMR in combination with multivariate analysis as a robust strategy to study complex matrices like bee propolis. This approach enables us to focus on critical compounds that play a key role in distinguishing differences among samples, allowing for a chemical classification. This is particularly significant when prior knowledge is limited, necessitating an exploratory study. 

The inactive compounds corresponded to agathadiol, previously isolated from Mediterranean propolis [31], and the *O*-methylated derivative of *n*-coniferyl benzoate (reported as a constituent of propolis for the first time). *n*-Coniferyl benzoate has been reported in Russian propolis [29] and has also been identified as a constituent of Peruvian balsam, used in the treatment of wounds and respiratory diseases. This balsam is obtained from the bark of *Myroxylom balsamum* (L.) [36]. Despite its presence in regions ranging from southern Mexico to the Brazilian Amazon and being reported in lowlands and altitudes up to 600 m above sea level [37], it is surprising that this component had not been previously reported in propolis from Mesoamerican regions. In contrast, nemorosone is widely distributed among *Clusia* spp., such as *C. rosea* (Jacq.), and has been identified in brown Cuban propolis [38]. It has also been found in propolis from Venezuela and Brazil [39].

These findings, which revealed the presence of three chemotypes of Costa Rican propolis, suggest that *Apis mellifera* bees tend to select resins from specific botanical sources in their surroundings more frequently than anticipated. Notably, the prevalence of nemorosone in apiaries from the South of Costa Rica, even at concentrations as high as 40–45% mass/mass (unpublished results), holds significant value, as recent anti-cancer research has shown its potential for treating this disease. For instance, nemorosone has demonstrated the ability to interfere with the growth of hepatocellular carcinoma cells mediated by macrophages, as evidenced by decreased colony numbers and cell migration at micromolar concentrations [40].

## 4. Materials and Methods

### 4.1. Reagents and Materials

Deuterated methanol (CD_3_OD, 98.9%), absolute ethanol, and analytical grade 2,2-diphenyl-1-pycrilhydrazyl (DPPH) were purchased from Sigma-Aldrich (St. Louis, MO, USA). HPLC (high Performance liquid chromatography)-grade methanol from Merck (Darmstadt, Germany) was used for the preparation of extracts. ACS quality gallic acid was obtained from Fluka (St. Louis, MO, USA) and maleic acid from Riedel de Haën (Seelze, Germany). SiO_2_ 60GF_254_ thin-layer chromatography (TLC, Merck, Darmstadt, Germany) plates, used in preparative (PTLC) separations, and SiO_2_ 60G and SiO_2_ 70–230 mesh (both Merck, Darmstadt, Germany), used for vacuum liquid (VLC) and flash column chromatography (CC), respectively, were obtained from Merck (Darmstadt, Germany). HPLC-grade solvents were used for HPLC, flash CC, VLC, and PTLC separations. HPLC separations were conducted using a SGE C18 (Wakosil II 5C18RS 250 × 4.6 mm (l × id, Wako Chemicals, Osaka, Japon) of 5 μm.

### 4.2. Equipment

Antioxidant quantitation was performed on an enzyme-linked immunosorbent assay (ELISA) reader (Thermo Scientific, Waltham, MA, USA). HPLC separations were conducted on a Shimadzu (Kyoto, Japan) chromatograph LC 20 with a refractive index as a detector. NMR measurements were conducted on a Varian Mercury 400 BB equipped with a 5 mm probehead (Palo Alto, CA, USA). Structures were obtained via NMR analysis using 1D and 2D experiments (^1^H, ^13^C, gCOSY, gHSQC, gHMBC, and NOESY) using the standard experiments provided by Varian. Molecular weights were obtained using an Acquity-Synapt Waters (Milford, MA, USA, liquid chromatograph mass spectometer quadrupole time of flight) LCMS qTOF, equipped with a Kinetex Phenomenex (Torrance, CA, USA) C18 column (50 × 2.1 mm) of 1.7 μm using a run of 3 min. Exact mass determinations were conducted on both negative and positive mode.

### 4.3. Sampling Strategy

Sampling was conducted using a randomized method without replacement, with a random number table [41] for selection. The table was applied on an updated database, with 2008 data from Costa Rican apiculturists kindly provided by the Ministry of Agriculture, which comprised information regarding 836 apiaries that produced honey and bee wax; of these, 24 apiaries were selected (2.87% of Costa Rican total apiaries). Actual collections occurred between September and October 2008. From each of the 24 apiaries, 5 hives were sampled in all but one, affording a total of 119 samples. All samples were transported on ice to the laboratory, where they were kept in a freezer at −15 °C until further use. For sample tracking, each apiary was given a number (1 to 24) and each hive within a given apiary was given a letter (A to E).

### 4.4. Preparation of Extracts

All propolis samples were lyophilized for 24 h. Samples were mortar pulverized and kept in vials in a freezer until further use. A 400 mg sample was taken from each propolis and mixed in 10.00 mL of methanol, then extracted using an ultrasonic bath for 15 min. Samples were centrifuged at 3000 rpm for 4 min at 24 °C, and the supernatants were transferred to sealed, labelled tubes and kept at 4 °C until further use.

### 4.5. Antioxidant Activity (AOA) Determination

DPPH scavenging reagent was used for the determination of antioxidant activities, as described by Gamez et al. [42], Cotelle et al. [43] and Mellors and Tappel [44]. Reaction mixtures containing test extracts were dissolved in ethanol and 100 μL of a DPPH solution (prepared using 8 mg of DPPH in 40 mL of ethanol 75% *v*/*v* and sonicated for 10 min) in 96-well microtiter plates, which were incubated for 15 min at 23 °C before the optical density was measured at 550 nm on an ELISA reader [45]. EC_50_ values (effective concentration to scavenge 50% DPPH from its original concentration) were calculated by plotting absorbance versus concentration. All samples were measured in duplicate. Gallic acid (366 ± 1 μg/mL) was used as a positive control. The results were expressed in µg of solids/mL of extract needed to reduce the original concentration of DPPH by 50%. Then, a correction factor was applied using the following formula:(1)$${EC}_{50}\;corrected={EC}_{50}\;measured\times extracted\;solids/original\;concentration$$

The evaporation of 1000 μL on a rotatory evaporator and subsequent drying to a constant weight in a vacuum oven at 40 °C afforded the extracted solids from each sample, to be used in Formula (1).

The DPPH assay of pure compounds was made in a similar way as before but starting from a solution of known concentration. The oxygen radical absorbance capacity (ORAC) values of pure compounds were quantified using a microplate assay based on the method described by Zamora et al. [46] without modifications.

### 4.6. Sample Preparation for ^1^H NMR Analysis

The extracts were dried in vials to produce samples weighing about 6.0 ± 0.1 mg. Then, a solution of 300 μL of CD_3_OD with maleic acid at a concentration of 16.70 ± 0.01 mg/mL as internal standard was added. The final solutions (20.0 ± 0.3 mg/mL) were transferred to Shigemi 5 mm tubes for NMR measurement.

The samples were measured on a Varian Mercury 400 MHz at 23 °C, using a 45° pulse (pulse width = 4.65 μs), a relaxation delay of 1.0 s, an acquisition time of 4.5 s, and an accumulation of 32 scans. The free induction decay (FID) size was 32 k data points, with a spectral width of 5995.2 Hz. Chemical shifts are given in ppm to one decimal point, and the spectra were centered using the deuterated solvent.

### 4.7. Statistical Analysis

Each FID was loaded into the MestReNova^®^ 14.21-27684 and referenced to the solvent signal at 3.31 ppm. The phase and baseline were manually adjusted, and all 119 FIDs were compiled in a single file. Regions between 3.26 and 3.33 (CD_2_HOD), 4.80and 5.00 (HOD), and 6.29and 6.31 ppm were left out of the analysis, as well as regions below 0.25 and above 10.25 ppm. The spectra were divided into bins of 0.04 ppm for a total of 210 bins. Each bin was normalized using a total sum of 1 and scaled using Pareto, following the PCA and GLR models in RStudio 1.3.1073 (running R version 4.0.2) and using the factoextra, ggrepel, vegan, ggpubr, car, dplyr, and ggplot2 packages. Metaboanalyst 5.0 was used to build the self-organizing map (SOM).

Complementary PCA and linear discriminant (DA) analysis based on the bins of isolated compounds as well as the PLS of the antioxidant activity and chemical fingerprints of samples with EC_50_ < 1000 µg/mL were conducted using Minitab^®^ 16.1.1 software on saved Microsoft Office^®^ Excel 2003 bins, where noise signals were deleted.

### 4.8. Isolation of Compounds

Lyophilized active propolis (6.9818 g of sample 6 A and 2.7680 g of sample 19 C) were extracted with 30 mL of MeOH for 24 h at RT. The following day, a final sonication of 10 min was employed. Each extract was centrifuged at 3000 rpm for 4 min and the supernatant was transferred to a dark vial. The extractions were repeated twice, using 20 mL of MeOH for each one. The final extracts were reduced under vacuum to yield 29.1 and 25.5 mg/mL for samples 6 A and 19 C, respectively. The separation of the active components was followed by TLC and DPPH as spray reagents (1 mg/mL in MeOH).

#### 4.8.1. Isolation of *n*-Coniferyl Benzoate (1)

VLC separation was performed on 500 mg of propolis extract labelled 6 A (75 g of Silica Gel for TLC on a büchner of 7 cm id) using hexane, which was followed by an increasing gradient of hexane:dichloromethane and dichloromethane:methanol mixtures, affording 13 fractions of 40 mL each. Fractions 7 and 8 (CH_2_Cl_2_:MeOH 99:1 *v*/*v*) were combined (72 mg) and a sample of 20 mg was further separated on a HPLC, using isocratic conditions *v*/*v* (MeOH:H_2_O 80:20 *v*/*v*) and a flow rate of 0.5 mL/min (1250 psi); the peak at 13.6 min was collected, affording 4.3 mg of *n*-coniferyl benzoate **1**. Identity of **1** was confirmed via NMR, LCMS, and comparison with data from the literature [30]. ^1^H NMR (400 MHz, CD_3_OD) ppm 8.05 (dd, *J* = 8.2, 1.4 Hz, 2H), 7.60 (tbr, *J* = 7.5 Hz, 1H), 7.49 (ddbr, *J* = 8.2, 7.5 Hz, 2H), 7.05 (d, *J* = 1.9 Hz, 1H), 6.90 (dd, *J* = 8.2, 2.0 Hz, 1H), 6.75 (d, *J* = 8.1 Hz, 1H), 6.69 (dt, *J* = 15.8, 1.3 Hz, 1H), 6.30 (dt, *J* = 15.8, 6.6 Hz, 1H), 4.95 (dd, *J* = 6.6, 1.2 Hz, 2H), 3.87 (s, 3H). ^13^C NMR (101 MHz, CD_3_OD) ppm 167.93 (C7′′), 149.14 (C3′), 148.06 (C4′), 136.01 (C3), 134.24 (C4′′), 131.56 (C1′′), 130.53 (C2′′, C6′′), 129.60 (C1′), 129.24 (C3′′, C5′′), 121.43 (C6′), 121.29 (C2), 116.22 (C5′), 110.61 (C2′), 67.08 (C1), 56.36 (C7′). HRMS: C_17_H_16_O_4_, detected as [M − H]^−^, 283.0968 (calc. mass 283.0970).

#### 4.8.2. Isolation of Nemorosone (2)

PTLC of 100 mg of propolis extract labelled 19C afforded three fractions (CHCl_3_:MeOH, 9:1 *v*/*v*). Fraction 2 (27.3 mg, R_f_ = 0.77) was further separated on HPLC (30 °C), using a mixture of MeOH:H_2_O (75:25 *v*/*v*) and a 0.80 mL/min flow rate (2032 psi), and the peak that eluted at Rt 10.2 min was collected; as such, 5.9 mg of nemorosone was recuperated. The identity of Fraction 2 was confirmed via NMR, LCMS, and comparison with data from the literature [1]. ^1^H NMR (400 MHz, CD_3_OD) ppm 7.78 (dbr, *J* = 8.6 Hz, 2H), 7.36 (tbr, *J* = 7.8 Hz, 1H), 7.20 (dd, *J* = 8.4, 7.8 Hz, 2H), 5.24 (tq, *J* = 7.0, 1.4 Hz, 1H), 5.11–4.99 (m, 2H), 3.06 (dbr, *J* = 6.5 Hz, 2H), 2.40 (dd, *J* = 6.6, 1.7 Hz, 2H), 2.13 (m, 1H), 2.01–1.90 (m, 2H), 1.66 (dd, *J* = 5.9, 1.3 Hz, 6H), 1.66–1.67 (m, 1H), 1.61 (sbr, 9H), 1.57 (sbr, 3H), 1.38 (s, 3H), 1.32–1.20 (m, 1H), 1.06 (s, 3H). ^13^C NMR (101 MHz, CD_3_OD) ppm 214.44 (C9), 197.91 (C10), 189.71 (C4), 187.19 (C2), 139.53 (C11), 133.12 (C24), 132.87 (C29), 132.35 (C14), 129.85 (C19), 129.72 (C12,C16), 128.31 (C13,C15), 126.16 (C18), 124.87 (C28), 123.04 (C23), 118.40 (C3), 79.48 (C1), 63.70 (C5), 47.44 (C8), 44.29 (C7), 42.42 (C6), 31.10 (C22), 28.66 (C27), 26.20 (C31, C26), 26.05 (C21), 23.53 (C17, C33), 18.31 (C20, C30), 18.25 (C25), 16.66(32). HRMS: C_33_H_42_O_4_ detected as [M − H]^−^, 501.3000 (calc. mass 501.3005) and [M + H]^+^, 503.3149 (calc. mass 503.3161).

#### 4.8.3. Isolation of DPPH Non-Active Compounds

Sample 23 C (80 mg) was separated into 4 fractions (La Potenciana in Turrubares) via preparative thin-layer chromatography (PTLC) using chromatographic plates of silica gel 60 GF_254_ and CHCl_3_:MeOH (95:5 *v*/*v*) as eluent. The fractions were codified as A (16.6 mg), B (10.7 mg), C (25.1 mg), and D (15.5 mg). Fraction C (9.5 mg) was injected onto a liquid chromatograph (Knauer 364) a with refractive index detector for the isolation of the most abundant compound, using a XTerra RP18column (150 × 3.9 mm × 5 µm of particle size) and a mixture of methanol:water (55:45 *v*/*v*). The flow was adjusted to 1.0 mL/min, with a pressure of 265 Kg/cm^2^. The separation was performed at room temperature (25 °C) with a peak attenuation of 16 to make its visualization on the employed scale easier. Each run took 20 min, and the peak eluted at 12.0 min was collected, which corresponded to agathadiol **3** (oil, 6.4 mg). This compound did not react with the DPPH reagent (methanolic solution of DPPH 1 mg/mL). The identify of **3** was confirmed via NMR, and comparison with data from the literature [31]. ^1^H NMR (ppm, CD_3_OD): 0.68 (3H, s), 0.94 (1H, m), 0.95 (3H, s), 1.07 (1H, dt, *J* = 8.8, 2.8Hz), 1.26 (1H, dd, *J* = 8.0, 1.2Hz), 1.32 (1H, ddd, *J* = 16.8, 8.8, 2.8 Hz), 1.46 (1H, m), 1.47 (1H, m), 1.57 (1H, m), 1.64 (1H, d, *J* = 6.8 Hz), 1.64 (1H, overlapped), 1.66 (3H, s), 1.81 (1H, m), 1.82 (1H, m), 1.82(1H, m), 1.87 (1H, m), 1.94 (1H, m), 2.14 (1H, m), 2.40 (1H, ddd, *J* = 8.4, 2.8, 1.6 Hz), 3.26 (1H, d, *J* = 7.2 Hz), 3.73 (1H, d, *J* = 7.2Hz), 4.08 (2H, d, *J* = 4.4 Hz), 4.52 (1H, d, *J* = 0.8 Hz), 4.83 (1H, d, *J* = 0.8Hz), 5.31 (1H, dt, *J* = 4.4, 0.8 Hz). ^13^C RMN (ppm, CD_3_OD): 149.6 (C8), 140.2 (C13), 124.7 (C14), 107.0 (C17), 64.8 (C19), 59.4 (C15), 57.6 (C5), 57.5 (C9), 40.5 (C10), 40.3 (C2), 39.9 (C4), 39.7 (C7), 39.4 (C6), 36.5 (C3), 27.8 (C18), 25.5 (C12), 23.0 (C11), 20.0 (C1), 16.3 (C16), 15.9 (C20).

A sample of 873 mg of 6 A extract (Nandayure, Guanacaste) was fractionated via flash chromatography (FC) with 120 g of Silica gel 70–230 mesh. The elution was performed with 250 mL CH_2_Cl_2_ (100%), 500 mL CH_2_Cl_2_ mixed with 50 drops MeOH, 2 × (230 mL CH_2_Cl_2_ mixed with 20 mL MeOH), and 500 mL MeOH (100%). Twenty-two fractions of nearly 80 mL each were collected and Fractions 8, 9, and 10 contained pure *O*-methyl coniferyl benzoate **4** (oil, 12.5 mg) (R_f_ = 0.51 on silica gel 60 GF_254_ CH_2_Cl_2_:MeOH (10 mL + 1 drop), detection UV = 254 nm). This compound did not react with the DPPH reagent (methanolic solution of DPPH 1 mg/mL). ^1^H NMR (ppm, CD_3_OD): 3.83 (3H, s), 3.85 (3H, s), 4.96 (2H, dd, *J* = 8.0, 1.2 Hz), 6.34 (1H, td, *J* = 15.6, 6.8 Hz), 6.71 (1H, d, *J* = 15.6 Hz), 6.90 (1H, d, *J* = 8.0Hz), 6.99 (1H, dd, *J* = 8.0, 2.0 Hz), 7.08 (1H, d, *J* = 2.0 Hz), 7.49 (2H, t, *J* = 8.0 Hz), 7.59 (1H, tt, *J* = 7.2, 1.2 Hz), 8.05 (2H, dd, *J* = 8.0, 1.2 Hz). ^13^C NMR (ppm, CD_3_OD): 167.9 (C7′′), 150.8 (C3′), 150.6 (C4′), 135.5 (C3), 134.2 (C4′′), 131.5 (C1′′), 131.1 (C1′), 130.5 (C2′′, C6′′), 129.6 (C3′′, C5′′), 122.3 (C2), 121.3 (C6′), 112.8 (C5′), 110.7 (C2′), 66.9 (C1), 56.4 (C7′), 56.4 (C8′).

## 5. Conclusions

In conclusion, this study revealed unexpected findings regarding the classification of Costa Rican propolis from *Apis mellifera*, based on its chemical composition, contrary to what we initially hypothesized. Costa Rican propolis can be classified into three chemotypes based on the composition of nemorosone, *n*-coniferyl benzoate, and terpene-rich extracts. The geographic zone emerged as a crucial determinant in defining the chemical type of a propolis sample, showcasing regional specificity. 

Remarkably, Chemotype II was found uniquely in the southern part of Costa Rica, while Chemotype I predominated in 45% of samples from Guanacaste. The prevalence of nemorosone in highly concentrated propolis samples is significant, considering recent studies that have investigated its potential in inhibiting the growth of cancer cells. Conversely, Chemotype III was prevalent in almost all regions except the Pacific Coast in the South. Utilizing multivariate techniques, we established a strong correlation between the antioxidant activity of propolis extracts and their chemical fingerprint, as measured via ^1^H NMR. Specifically, extracts with highly intensive signals of nemorosone and *n*-coniferyl benzoate demonstrated superior antioxidant activity. 

Based on these findings, we concluded that these compounds serve as useful chemical markers for categorizing Costa Rican bee propolis samples into distinct classes. Additionally, these compounds can be used as quality standards, thereby enhancing the overall value of bee propolis. The combination of ^1^H NMR and multivariate analysis has proven to be valuable in discriminating Costa Rican bee propolis and predicting their antioxidant activity based on their chemical profiles.

It is important to highlight that these results depict a moment in time and place. Further studies conducted in the southern regions have consistently showed the occurrence of nemorosone in that area (unpublished results). However, if drastic changes occur in the surrounding flora, propolis compositions could change, making subsequent studies of the composition relevant and necessary.

## Figures and Tables

**Figure 1 molecules-28-07081-f001:**
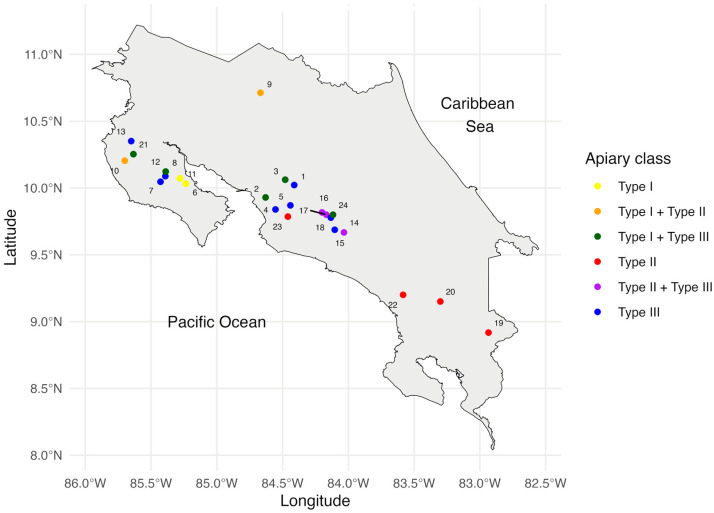
Apiaries selected for this study and their geographical distribution. Zones are highlighted and labelled according to chemotype (color codes). All of them were GPS georeferenced.

**Figure 2 molecules-28-07081-f002:**
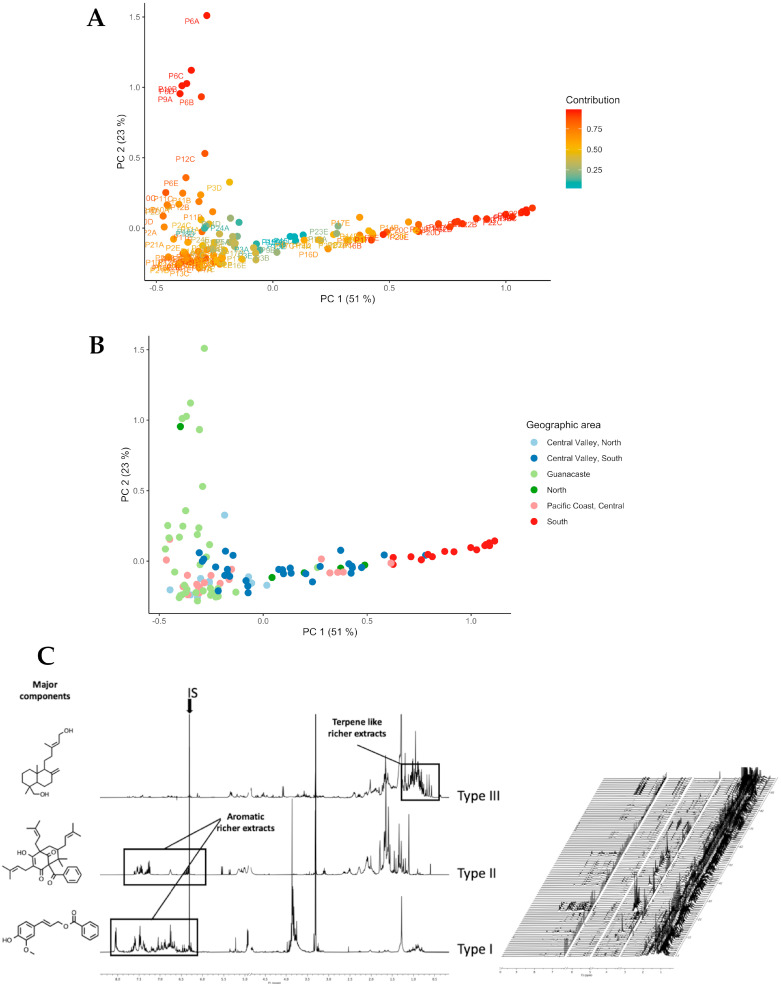
(**A**) PCA score plot obtained with 0.04 ppm thick bins going from 0.3 upfield to 9.5 ppm downfield. Signals from the internal standard, the water, and the residual solvent were deleted when obtaining the normalized, scaled table. (**B**) PCA score plot, with points colored according to origin. (**C**) On the right, a stack of the ^1^H NMR spectra of 119 extracts; on the left, the ^1^H NMR representative spectra of samples originating from the three distinct clusters that emerged after SOM analysis.

**Figure 3 molecules-28-07081-f003:**
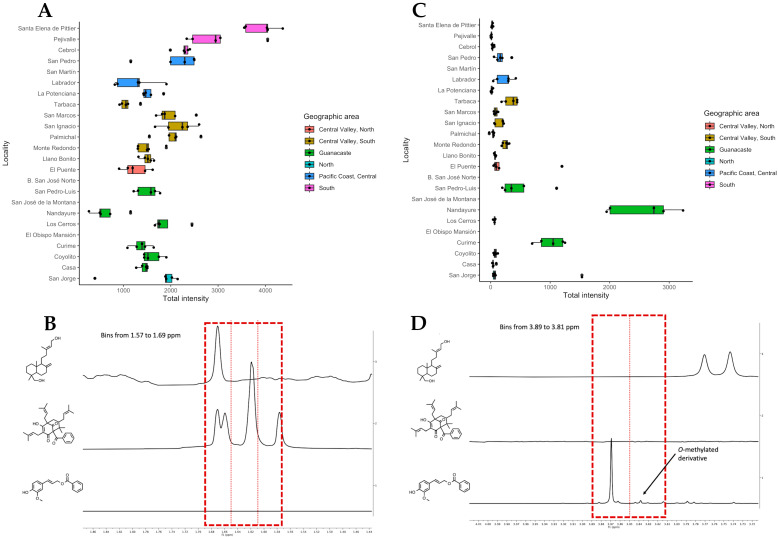
(**A**) Contribution weights of bins from 1.57 to 1.69 ppm in the PCA analysis. (**B**) Corresponding NMR signals of pure isolated compounds. As can be observed from the boxplot, samples with higher content on those bins were related to the South and to the presence of nemorosone. (**C**) Contribution weights of bins from 3.81 to 3.89 ppm in the PCA analysis. (**D**) Corresponding NMR signals from pure isolated compounds. As can be seen, the methoxy groups of the two coniferyl benzoates appeared in that region and were found mainly in propolis from Guanacaste.

**Figure 4 molecules-28-07081-f004:**
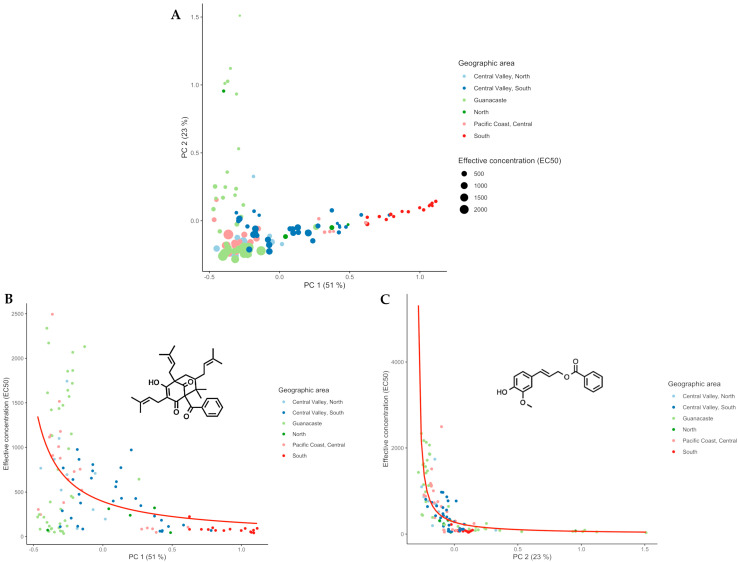
(**A**) PCA score plot labelled according to origin and EC_50_ DPPH antioxidant test results, where the size of a circle is related to the effect observed: the larger the circle, the less antioxidant the extract is; the smaller, the more effective. (**B**) GLM on PC1, where the most important extracts were the ones from the South and related to nemorosone **2**. (**C**) GLM on PC2, where extracts from Guanacaste were the ones found to be more effective and related to *n*-coniferyl benzoate **1**.

## Data Availability

Supporting information can be accessed as described above. Further NMR experiments can be provided upon request.

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
