# Peer review of "Costa Rican Propolis Chemical Compositions: Nemorosone Found to Be Present in an Exclusive Geographical Zone"

_molecules, 2023, doi:10.3390/molecules28207081_

Round 1
Reviewer 1 Report
The main question addressed by the research is the variation in composition of Costa Rican propolis. The topic is relevant in the field, and this country`s propolis in not fully studied. Compared with other published material, More information on propolis from Costa Rica is added to the subject area. The references are appropriate.
English editing is necessary throughout.
It is necessary to cite the bee species in the title (if possible), in the abstract, material and introduction. The species (Apis mellifera) was only cited in line 339 of the text!
Do the three chemotypes correlate with specific environments? This would enrich the discussion.
English editing is necessary throughout . At the beginning I listed the corrections , but gave up after a while. See below:
Corrections:
Line 99-find cultivars – of what?
104- in a fridge or freezer?
113 and 116- apiarian cultivars?? Apiaries is more usual.
119 – change to – most samples from Guanacaste…
121- as can be seen in Figure..
135- can be grouped in..
204- twenty chemically different carbons
337 concentrations not exceding
Review English throughout!!!
Author Response
Reviewer #1: Changes are highlighted in yellow in the draft. Thanks for all the observations and suggestions!!
|
# |
Reviewer comment |
Authors response |
|
1 |
English editing is necessary throughout.
|
We did our best effort to revise the whole manuscript and corrected grammatical errors as well as some styling mistakes. |
|
2 |
It is necessary to cite the bee species in the title (if possible), in the abstract, material and introduction. The species (Apis mellifera) was only cited in line 339 of the text! |
Done! Thanks for the observation! |
|
3 |
Do the three chemotypes correlate with specific environments? This would enrich the discussion.
|
Presently we don’t have thorough information on the environments surrounding apiaries. Sorry about this! |
|
4 |
English editing is necessary throughout .
|
Again, we revised the whole document and made changes accordingly.
|
|
5 |
Line 99-find cultivars – of what?
|
“cultivars” was changed by apiary/apiaries accordingly. Thanks for the observation. |
|
6 |
104- in a fridge or freezer?
|
Corrected! Freezer. |
|
7 |
113 and 116- apiarian cultivars?? Apiaries is more usual.
|
See above. Changed! |
|
8 |
119 – change to – most samples from Guanacaste…
|
Revised and changed. |
|
9 |
121- as can be seen in Figure..
|
Changed! Thanks for the observation. |
|
10 |
135- can be grouped in..
|
Changed to clusters or clustering or clustered. Thanks! |
|
11 |
337 concentrations not exceding
|
I could not find this correction. Probably I changed it when editing? Hope so! |
Reviewer 2 Report
The manuscript is interesting. I have some remarks and recommendation that can improve it.
1. The bee species whose propolis samples are examined is only mentioned in the Conclusion section. Add it right at the beginning of the manuscript.
2. The Abstract should be improved by clearing the way of expression especially in the parts of Results and Conclusions.
3. I recommend some previous studies/data about chemical composition of propolis originating from Central American region also to be mentioned in the Introduction section.
4. Change 1H NMR to 1H NMR as well as EC50 to EC50.
5. The designation "δppm" is incorrect. Use only "δ" or "ppm". For example: 3.85 δppm have to be δ 3.85 or 3.85 ppm. Change it in the whole text.
6. In many places in the text, the term "cultivar" is used incorrectly. Remove it or replace it with "sample".
7. Check the References (17-20). Some redundant numbers left. No 18 is from No 17, so the following numbers need to be fixed.
Some grammatical and stylistic error.
Author Response
|
# |
Reviewer comment |
Authors response |
|
1 |
“The bee species whose propolis samples are examined is only mentioned in the Conclusion section. Add it right at the beginning of the manuscript.”
|
Thanks for this observation! We added at the beginning and mentioned it afterwards. |
|
2 |
“The Abstract should be improved by clearing the way of expression especially in the parts of Results and Conclusions. |
We hope to have improved the abstract accordingly, although the permitted number of words is quite limiting. |
|
3 |
“I recommend some previous studies/data about chemical composition of propolis originating from Central American region also to be mentioned in the Introduction section.”
|
Studies from Honduras and El Salvador were mentioned in the Introduction. Thanks for the observation! |
|
4 |
“Change 1H NMR to 1H NMR as well as EC50 to EC50.”
|
Thanks! We changed it accordingly. |
|
5 |
“The designation "δppm" is incorrect. Use only "δ" or "ppm". For example: 3.85 δppm have to be δ 3.85 or 3.85 ppm. Change it in the whole text.”
|
Thanks! We changed it accordingly. |
|
6 |
“In many places in the text, the term "cultivar" is used incorrectly. Remove it or replace it with "sample".” |
Thanks for this observation. It was replaced by apiary or sample throughout the text. |
|
7 |
“Check the References (17-20). Some redundant numbers left. No 18 is from No 17, so the following numbers need to be fixed.” |
We revised all references and made the changes accordingly. Thanks for the observation. |
|
Final comment on quality of English Language |
“Some grammatical and stylistic error.” |
We did our best effort to revise the whole manuscript and correct grammatical errors as well as some styling mistakes. |